# Integrating Segmented Cell Imaging and Molecular Networks for Drug-Specific Analysis in CM4AI

## Abstract

Linking cellular morphology to molecular interaction networks remains a central challenge in biomedical Artificial Intelligent (AI). We present a cell-centric framework that integrates object-level detection with Vision Transformer (ViT) embeddings from microscopy with protein–protein interaction (PPI) representations to construct biologically interpretable hierarchies and reveal condition-specific network reconfiguration. Using a semi-automated, agent-oriented workflow, segmentation is executed via an interactive Large Language Model (LLM)-driven agent bridged to high-performance computing, while embedding, integration, and hierarchy construction proceed through reproducible human–LLM collaboration with auditable prompts, code generation, and logged execution. Applied to 12853 high-content images spanning Untreated, Vorinostat-, and Paclitaxel-treated conditions, the approach preserves global biological structure while sharpening signal fidelity relative to whole-image baselines, enabling single-cell resolution of heterogeneity. Across all conditions, the modified pipeline maintained >95% concordance with baseline hierarchies. Gene Ontology analyses recover drug-consistent pathways (e.g., chromatin regulation for Vorinostat; microtubule-associated processes for Paclitaxel) and yield more selective enrichment profiles. The framework establishes a scalable foundation for multimodal integration with additional omics layers and for prospective validation of predicted network rewiring in precision medicine contexts.

## 1 Introduction

Linking cellular morphology to molecular interaction networks is a central challenge in systems biology and drug discovery [1, 2]. High-content microscopy provides rich phenotypic readouts [3-5], while protein–protein interaction (PPI) networks encode molecular relationships that govern function [1, 2, 6]. Yet, these modalities are typically analyzed in isolation, limiting our ability to explain how drug-induced morphological changes propagate to network-level rewiring [7, 8]. Overcoming this gap is critical for uncovering mechanisms of action and advancing precision therapeutics [9, 10].

This problem presents three significant machine learning challenges with important consequences: (1) **Data heterogeneity**—images and graphs differ fundamentally in structure, making cross-domain representation learning difficult. Without effective integration, morphology–network correspondences are often missed or distorted [11-13]; (2) **Scalability and efficiency**—microscopy datasets contain millions of cells, yet whole-image embeddings are dominated by background regions. This wastes computation, inflates storage, and introduces noise, leading to reduced predictive accuracy [14, 15]; and (3) **Interpretability**—without transparent mapping between morphology and molecular pathways, models risk becoming black boxes, limiting biological trust and practical utility [16-17].

Existing approaches address these challenges only partially and can be grouped into three categories. **Aggregation-based methods** such as MuSIC fuse imaging with molecular data but collapse features

across cell populations, losing single-cell heterogeneity [7]. **Static resources** like OpenCell provide valuable maps of localization and interaction but are not designed as dynamic learning frameworks [8]. **Morphology profiling assays** such as Cell Painting and its large-scale derivatives capture phenotypic diversity but do not connect directly to molecular interaction networks [8, 18]. Finally, **multimodal contrastive methods** (e.g., MaxFuse, MoCoP) highlight the potential of aligning weakly linked modalities, but they have not been applied to morphology–PPI integration and do not resolve the inefficiency of whole-image embeddings [19, 20].

We participated in the NIH-funded Cell Maps for Artificial Intelligence (CM4AI) initiative, which was established to link cellular morphology with PPI networks [21]. While CM4AI demonstrated the promise of multimodal integration, it primarily operated on whole-image embeddings, where large portions of each microscopy image consist of empty background or non-informative regions. Treating entire fields of view as single units forced the model to allocate capacity to irrelevant pixels, diluting the signal from actual cellular structures. Moreover, features were aggregated at the image or population level, which masked single-cell variability that is critical for capturing heterogeneity in drug response. As a result, background noise reduced predictive accuracy, and averaging across populations obscured subtle but biologically important differences between individual cells.

Our contributions are as follows:

- **Object-centric embedding pipeline:** A segmentation and embedding pipeline that reduces background noise and improves the signal-to-noise ratio, leading to more faithful representations of cellular morphology and improved downstream predictive accuracy of morphology–PPI alignment.
- **Agent-based orchestration:** We develop a semi-automated agent architecture where (Large Language Model) LLMs generate, refine, and execute code in collaboration with human researchers. This orchestration—detailed in the method section—establishes a reproducible workflow that goes beyond simple chaining of off-the-shelf components and represents a methodological advance for AI-driven scientific pipelines.
- **Cross-domain alignment:** We present novel preprocessing, mapping, and validation procedures for integrating morphology and network embeddings, including ontology-based evaluation with Gene Ontology (GO) analysis.

The rest of this article is organized as follows. Section 2 details the proposed methodology, beginning with the multi-agent architecture and then describing algorithms for each pipeline stage: object detection and segmentation, feature extraction, PPI embedding, multimodal integration, and hierarchy generation. Section 3 presents results on drug-specific datasets, including baseline vs. modified pipelines, network analyses, and Gene Ontology/KEGG enrichment. Section 4 examines biological and computational implications, limitations, and future directions. Finally, Section 5 concludes and highlights broader applications in systems biology and precision medicine.

## 2 Proposed Methods

### 2.1 Multi-Agent Architecture Implementation for CM4AI

The CM4AI pipeline proceeds through seven auditable stages for integrating cellular morphology with molecular interaction networks: (1) experimental condition parsing and dataset registration, (2) microscopy image ingestion and pre-processing, (3) object detection and LLM-guided segmentation, (4) feature extraction using Vision Transformers (ViT) at the single-cell level, (5) protein–protein interaction (PPI) embedding generation via graph-based models, (6) multimodal contrastive co-embedding with alignment scoring, and (7) hierarchical map construction with ontology-based validation. Execution was coordinated manually by the researchers with LLM assistance rather than by a centralized orchestrator: stage (3) used an interactive agent via a web interface bridged to the High-Performance Computing (HPC) system, whereas stages (4)–(7) were run from user-invoked scripts generated or refined by ChatGPT (and, where noted, Gemini). Transparent logging, metadata propagation, and reproducibility tracking were implemented through scripts and notebooks (e.g., recorded prompts/configurations, fixed seeds, saved checkpoints), ensuring each step is auditable across imaging and network data modalities.

Table 1 highlights the balance of human and AI involvement across the CM4AI pipeline.

a-Manual execution of pre-existing scripts written by human researchers.b-No code involved; task consisted of visual/manual inspection of generated networks.

Table 1: Human vs. AI involvement per pipeline stage.

| Stg | Activity | AI code (%) | Human code (%) | Executor Manual (%) | Executor Agent (%) |
|-----|----------|-------------|----------------|---------------------|---------------------|
| 1 | Planning | ∼80 | ∼20 | 100 | 0 |
| 2 | Manuscript | ∼80 | ∼20 | 100 | 0 |
| 3 | Detection & Segmentation | ∼95 | ∼5 | 30 | 70 |
| 4 | ViT features extraction | ∼95 | ∼5 | 100 | 0 |
| 5 | PPI embedding & Co-embedding[a] | 0 | 100 | 100 | 0 |
| 6 | Hierarchy generation | 50 | 50 | 100 | 0 |
| 7 | Ontology check[b] | 0 | 0 | 100 | 0 |

### 2.1.1 PlanningAgent

The PlanningAgent was used to outline the overall research workflow and formulate experimental ideas. Through prompt-driven instructions, ChatGPT proposed stepwise pipelines, suggested alternative methodological options, and generated schematic drafts of module interactions. Beyond technical planning, ChatGPT was also used to brainstorm research directions, especially to identify innovative approaches for emerging fields. For example, we prompted ChatGPT with the request: *"You are an expert in drug discovery and computer vision. Propose an approach that could achieve a breakthrough."*, which generated candidate directions that were later refined. Some preliminary studies were planned and carried out during this stage before the framework reached its current milestone. These outputs were not executed directly but served as planning material that the user evaluated, modified, and selected for implementation. Thus, the agent functioned as an ideation and organizational support tool rather than an autonomous decision-maker, with final choices and experimental designs determined by the human researcher.

### 2.1.2 ManuscriptAgent

The ManuscriptAgent supported manuscript preparation by generating draft text, structuring sections in LATEX, suggesting titles and figure captions, and providing language refinement. While ChatGPT produced most of the drafting and formatting support, some parts (e.g.: Introduction and Proposed Method) of the paper were written using Gemini through its agent API, which enabled iterative refinement of specific sections via repeated prompt–response cycles. The user remained responsible for verifying content accuracy, correcting hallucinated references, and ensuring logical consistency. In practice, the ManuscriptAgent accelerated drafting and improved readability, but the final paper required substantial human oversight and revision, reflecting a collaborative human–LLM writing process rather than a fully automated system.

### 2.1.3 SegmentationAgent

We first prompted ChatGPT to perform cell segmentation on the images, and then requested the corresponding source code. For instance, we used prompts such as: *"You are an expert in the field of biomedical image analysis. Provide bounding boxes for the cells in this image."* Next, we executed segmentation of all images through a ChatGPT-provided agent workspace running in a web interface bridged to our institute's HPC server, using stepwise, prompt-driven instructions. At each step, we supplied the agent with specifications (dataset layout, channel usage, output schema, and test criteria), and the agent generated environment setup scripts (conda/pip with PyTorch, OpenCV, and other libraries), and executable code for object localization. After pilot verification on sample images, the agent launched batched jobs across the full dataset, streamed logs/metrics, and wrote masks, boxes, and cell crops with provenance and metadata. When failures occurred, such as technical errors (e.g.,

missing files, broken dependencies) or operational mistakes (e.g., the agent clicking an incorrect button and failing to correct the action in the web interface), the user had to manually take control of the screen to recover and continue execution. Thus, the system operated as a human–LLM co-pilot rather than a fully autonomous agent.

### 2.1.4 EmbeddingAgent

The EmbeddingAgent was responsible for generating feature representations from both cell images and protein interaction data. For cell morphology, a ViT backbone was used to extract features, relying on a combination of existing libraries and code generated by ChatGPT in response to prompt instructions. Unlike the segmentation stage, this module was not executed through an autonomous agent interface; instead, ChatGPT produced scripts that the user manually executed on the HPC server. Some preprocessing was handled manually, and errors in processing occasionally required refinement of the code or reruns. Thus, while ChatGPT assisted in code generation and troubleshooting, the embedding stage remained largely user-driven rather than agent-operated.

### 2.1.5 CoEmbeddingAgent and HierarchyAgent

The CoEmbeddingAgent was responsible for aligning image-derived embeddings with protein interaction embeddings, while the HierarchyAgent organized the integrated features into higher-level biological groupings. In both cases, previously generated scripts were reused and adapted with minor modifications based on new prompt instructions, rather than written entirely from scratch. These scripts were executed manually on the HPC, with ChatGPT providing refinements and troubleshooting support, but without reliance on a fully autonomous interface.

## 2.2 Modified Pipeline

### 2.2.1 Input Data

We constructed a dataset $\mathcal{D}$ composed of approximately 12853 microscopy images:

$$\mathcal{D} = \{(I_i, c_i)\}_{i=1}^{N}, \quad N \approx 12853 \tag{1}$$

where $I_i \in \mathbb{R}^{H \times W \times C}$ is a high-resolution fluorescence microscopy image and $c_i \in \{\text{Untreated}, \text{Paclitaxel}, \text{Vorinostat}\}$ denotes the experimental condition.

### 2.2.2 Object Detection and Segmentation.

Raw images contain multiple nuclei, and global embeddings risk averaging out informative local variations. To address this, we applied an LLM-guided object detection function $f_{\text{det}}$ that isolates nuclei at the single-cell level, $\mathcal{S}_i = f_{\text{det}}(I_i) = \{s_{i1}, s_{i2}, \ldots, s_{iM_i}\}$, where $M_i$ is the number of detected objects in $I_i$ and each segment $s_{ij}$ corresponds to a nucleus or cell mask. This step preserves subtle morphological features, such as nuclear size and chromatin condensation, that are often lost at the global scale.

### 2.2.3 Feature Extraction.

Each segment $s_{ij}$ is embedded using a Vision Transformer (ViT) as $\mathbf{z}_{ij}^{\text{img}} = f_{\text{ViT}}(s_{ij}; \theta_{\text{ViT}}) \in \mathbb{R}^{d_{\text{img}}}$, where $\theta_{\text{ViT}}$ are the learned parameters. The ViT partitions $s_{ij}$ into patches, encodes them as tokens, and applies multi-head self-attention to capture both local and global dependencies.

## 2.3 PPI Embedding

To incorporate molecular context, we constructed a protein–protein interaction (PPI) graph $G = (V, E)$, where $V$ denotes proteins and $E$ their interactions. Using a graph embedding function $f_{\text{PPI}}$, each protein $p \in V$ was mapped to a vector $\mathbf{z}_p^{\text{ppi}} = f_{\text{PPI}}(G; p) \in \mathbb{R}^{d_{\text{ppi}}}$. These embeddings preserve both local and global network structure, such that inner products $\langle \mathbf{z}_u^{\text{ppi}}, \mathbf{z}_v^{\text{ppi}} \rangle$ reflect the likelihood of interaction between proteins $u$ and $v$.

### 2.4 Co-Embedding (Multimodal Integration)

To integrate protein image and PPI embeddings into a shared space, we applied the MUSE (*Multimodal Unsupervised Semantic Embedding*) framework. Let $\mathbf{x}_i \in \mathbb{R}^{d_x}$ and $\mathbf{y}_i \in \mathbb{R}^{d_y}$ denote the image and PPI embeddings for protein $i$ in the intersection set $\mathcal{I}$, which are mapped into a common latent space of dimension $d$ via neural functions $f_\theta$ and $g_\phi$, yielding $\mathbf{z}_i^{(x)} = f_\theta(\mathbf{x}_i)$ and $\mathbf{z}_i^{(y)} = g_\phi(\mathbf{y}_i)$. For each anchor $i$, the positive pair $(\mathbf{z}_i^{(x)}, \mathbf{z}_i^{(y)})$ is contrasted against negatives $(\mathbf{z}_i^{(x)}, \mathbf{z}_j^{(y)})$ from the $k$ nearest neighbors $(j \neq i)$ using a triplet margin loss $\ell_i = \max\left(0, \|\mathbf{z}_i^{(x)} - \mathbf{z}_i^{(y)}\|_2^2 - \|\mathbf{z}_i^{(x)} - \mathbf{z}_j^{(y)}\|_2^2 + m\right)$, where $m$ is the margin. The overall objective $\mathcal{L}_{\mathrm{MUSE}} = \sum_{i \in \mathcal{I}} \ell_i + \lambda \Omega(\theta, \phi)$ includes dropout and L2 regularization, and training proceeds in two stages: initialization for $n_{\mathrm{init}}$ epochs followed by triplet-based optimization for $n_{\mathrm{epochs}}$ epochs. This process aligns image and PPI features of the same protein while enforcing separation from unrelated proteins.

### 2.5 Hierarchy Generation

The joint embedding matrix was defined as $Z = [\mathbf{z}_{ij}^{\mathrm{joint}}] \in \mathbb{R}^{M \times d}$, and pairwise cosine similarities were computed as $\mathrm{sim}(\mathbf{z}_a, \mathbf{z}_b) = \frac{\mathbf{z}_a \cdot \mathbf{z}_b}{\|\mathbf{z}_a\|\|\mathbf{z}_b\|}$. Hierarchical agglomerative clustering was then applied using Ward's method, where the linkage between clusters $A$ and $B$ is given by $\Delta(A, B) = \frac{|A| \cdot |B|}{|A| + |B|} \|\mu_A - \mu_B\|^2$, with $\mu_A$ and $\mu_B$ denoting the respective centroids.

## 3 Results

### 3.1 Dataset

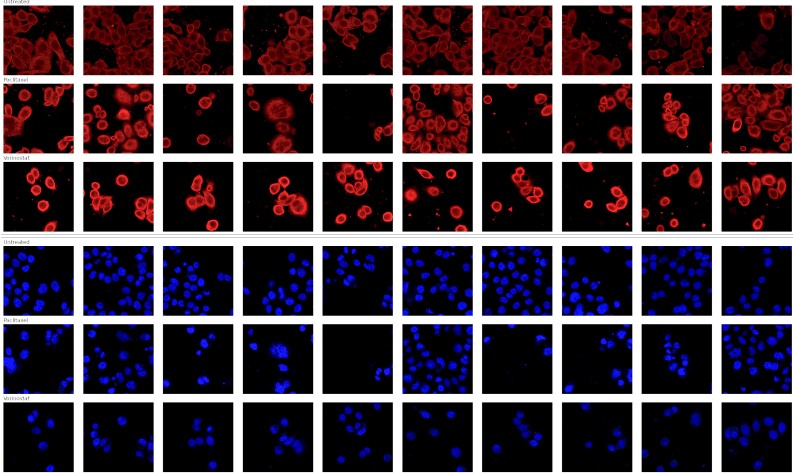

Figure 1: Representative microscopy images from the CM4AI dataset across three experimental conditions: Untreated, Paclitaxel-treated, and Vorinostat-treated.

The foundation of our analysis is the publicly available Cell Maps for Artificial Intelligence (CM4AI) dataset, produced under the NIH Bridge2AI Functional Genomics Grand Challenge [21]. As of the June 2025 beta release, CM4AI provides AI-ready RO-Crate archives with rich provenance via the FAIRSCAPE framework.

Our study uses the subset of immunofluorescent images of MDA-MB-468 breast cancer cells under three conditions—Untreated, Paclitaxel, and Vorinostat (Fig. 1). Each condition includes microtubule and nuclear channels, which we use for object detection and ViT embedding. CM4AI targets a curated panel of 200 human proteins (100 chromatin modifiers, 100 metabolic enzymes) selected for relevance to cancer, neuropsychiatric, and cardiac disorders, and also provides additional modalities (e.g., proteomics, perturb-seq, spatial proteomics) packaged with interoperable metadata under FAIR principles.

## 3.2 Hyper-parameter

Our proposed methods were developed using the `PyTorch`™ framework and implemented on a HPC running a Red Hat® Enterprise Linux 7 system equipped with two NVIDIA® Tesla™ P100-PCIE-16GB GPUs and 8 CPU cores (16 GB each) (Table 2). For details of the original pipeline settings, please refer to a previous publication [21].

Table 2: Experimental settings and hyperparameters.

| Module | Parameter | Value |
|---|---|---|
| *Segmentation* | | |
| | GAUSS SIGMA | 1.2 |
| | OPEN FOOT | 1 |
| | CLOSE FOOT | 3 |
| | MIN OBJ FRAC | 0.0003 |
| | MIN HOLE FRAC | 0.0006 |
| *ViT (image features)* | | |
| | Backbone | ViT large patch16 224 |
| | Image input size | 224 |
| | Feature dim $d_{\mathrm{img}}$ | 1024 |
| *DenseNet (image features)* | | |
| | Backbone | DenseNet 121 |
| | Image input size | 224 |
| | Feature dim $d_{\mathrm{img}}$ | 1024 |
| *PPI (graph features)* | | |
| | Embedding method | Node2Vec |
| | Feature dim $d_{\mathrm{ppi}}$ | 1024 |
| *Co-Embedding (MUSE)* | | |
| | Shared dim $d$ | 128 |
| | Margin $m$ | 0.1 |
| | Negatives / neighborhood $k$ | 10 |
| | Init epochs $n_{\mathrm{init}}$ | 200 |
| | Total epochs $n_{\mathrm{epochs}}$ | 500 |
| *Compute / Environment* | | |
| | GPUs / CPUs / RAM | 2/8/128 |
| | Runtime | 1 month |
| | Env hash | conda |
| *Reproducibility* | | |
| | Random seeds | 42 |

## 3.3 Comparison of Baseline and modified CellMaps Pipelines

To evaluate the consistency of our modifications, we compared protein hierarchies generated by the baseline CellMaps pipeline and the Modified CellMaps pipeline across untreated and drug-treated conditions (Fig. 2). The baseline approach derives hierarchies from global image embeddings, whereas the Modified pipeline incorporates object detection and ViT-based single-cell embeddings.

For the untreated condition (Fig. 2a), the overlap between the two approaches was high: 95 proteins were shared (97.9% concordance), with only one protein (RACK1) unique to the baseline hierarchy and one (TUBB8) to the Modified hierarchy.

For Paclitaxel (Fig. 2b), 92 proteins were shared (96.9% overlap). Four proteins (CBX3, RPS3, SMARCA5, ZBTB7B) appeared only in the baseline pipeline, while one (SMARCA4) was unique to the Modified version. This indicates that the Modified pipeline preserves Paclitaxel-associated modules but slightly reshapes centrality and regulatory protein membership.

For Vorinostat (Fig. 2c), 93 proteins overlapped (95.9% concordance). Four proteins (CPT1A, DNMT1, DNMT3A, SRP14) were unique to the baseline pipeline, and none to the Modified version, suggesting that the Modified approach retains nearly all Vorinostat-associated proteins while simplifying the hierarchy.

In addition, we also performed comparison of the original pipeline and modified pipeline of two settings (Paclitaxel and Vorinostat). DenseNet/whole-image approach summarizes entire fields of view, which dilutes cell-level signal and tends to produce more centralized network modules and

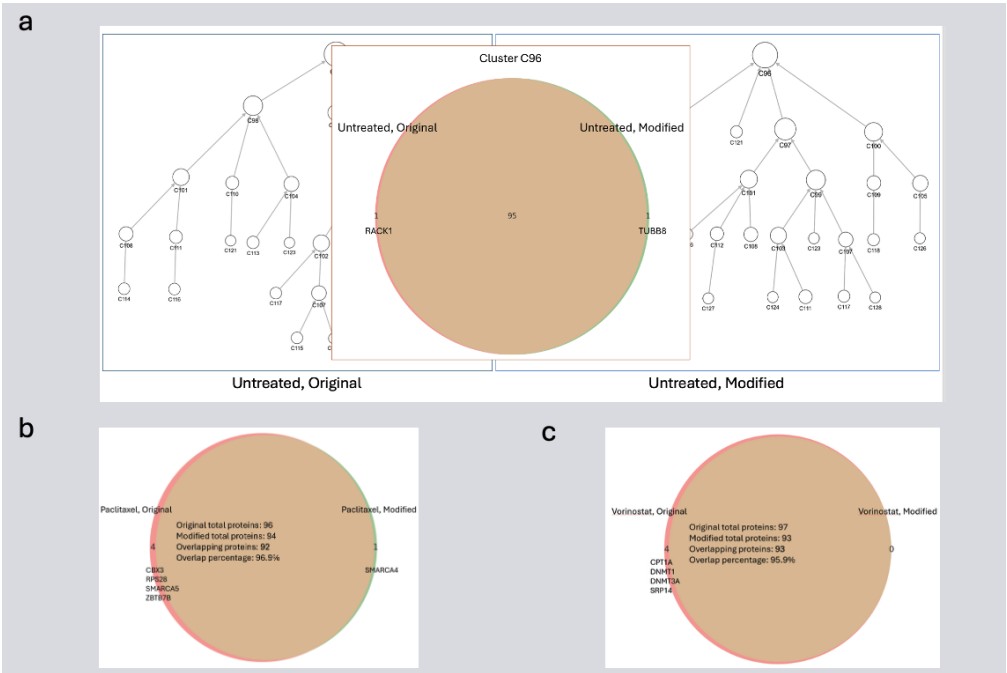

Figure 2: Comparison of baseline (Original) and Modified CellMaps hierarchies across conditions. (a) Untreated: 95 shared proteins (97.9% overlap), 1 unique to each pipeline. (b) Paclitaxel: 92 shared proteins (96.9% overlap), 4 unique to baseline, 1 unique to Modified. (c) Vorinostat: 93 shared proteins (95.9% overlap), 4 unique to baseline, none unique to Modified.

broader GO categories. The modified ViT/object-centric pipeline embeds individual cells, raising signal-to-noise, preserving heterogeneity, and yielding distributed hierarchies with more selective enrichment—i.e., it refines rather than replaces the original organization.

### 3.4 Drug-specific molecular reprogramming captured by the modified pipeline

We compared the modified networks for Paclitaxel and Vorinostat to resolve treatment-specific effects (Fig. 3). Both conditions preserved core chromatin and metabolic signatures, including strong enrichment in ATP-dependent chromatin remodeling (adenosine triphosphate-dependent), fatty acid metabolism, and nuclear components, confirming the stability of the modified pipeline across perturbations.

Distinct profiles emerged with each drug. Paclitaxel uniquely amplified pathways linked to long-term potentiation and oocyte meiosis, consistent with drug-induced disruption of signaling and cell cycle regulation. Vorinostat, in contrast, showed selective enrichment in the spliceosome and amino acid biosynthesis, aligning with its mechanism as a histone deacetylase inhibitor that broadly impacts transcriptional and RNA processing. These results highlight the pipeline's capacity to preserve shared biological modules while sensitively resolving drug-specific molecular reprogramming.

## 4 Discussion

We present a cell-centric framework that links single-cell morphological embeddings to PPI features, capturing heterogeneity while preserving baseline structure. High concordance (97.9% overlap) and more selective GO enrichment show that object-level embeddings reduce noise and sharpen signal without loss of fidelity.

Methodologically, our contribution is not only a change in representation but also a transparent, agent-supported workflow: segmentation is executed via an interactive agent bridged to HPC, and embedding/integration/hierarchy steps proceed through audited, human-in-the-loop code generation

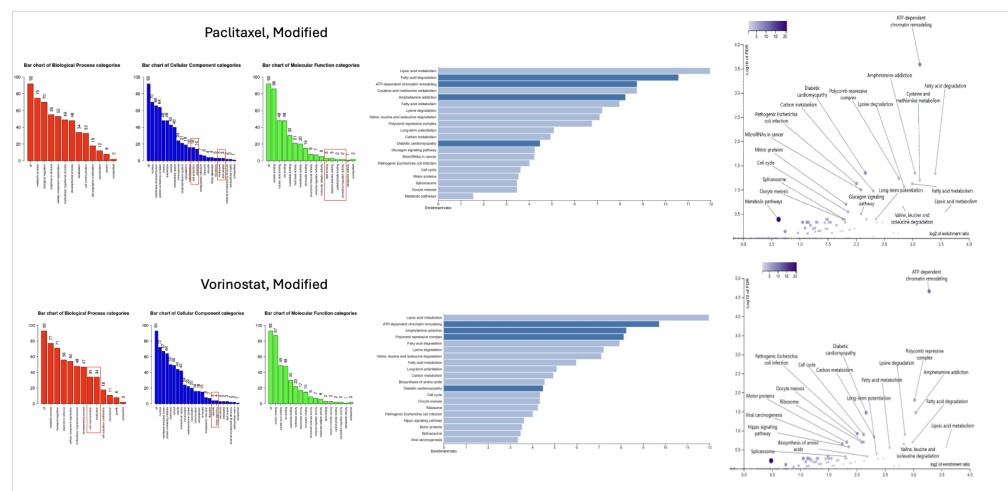

Figure 3: Functional enrichment profiles of Paclitaxel- and Vorinostat-modified networks. Bar plots (left) summarize Gene Ontology categories across biological process, cellular component, and molecular function. Enrichment ratios (center) and pathway significance plots (right) highlight shared chromatin and metabolic signatures while resolving drug-specific differences. Paclitaxel uniquely enriched long-term potentiation and oocyte meiosis, whereas Vorinostat emphasized the spliceosome and amino acid biosynthesis.

and execution. This orchestration emphasizes provenance, failure handling, and reproducibility, offering a practical template for agentic scientific computing in multimodal biology.

Relative to image-profiling pipelines that rely on whole-image aggregation or handcrafted features, the object-level strategy improves signal-to-noise and reduces spurious enrichments while maintaining global coherence. Unlike general cross-modal alignment methods, our framework targets morphology–PPI integration and operationalizes ontology-grounded evaluation within a unified workflow.

Our approach has several limitations. Segmentation metrics were not reported due to lack of ground truth in CM4AI, so evaluation relied on visual inspection. We also rely on ViT backbones not pretrained on fluorescence microscopy, which may miss domain-specific cues; domain-adapted pretraining could improve sensitivity. Enrichment analyses remain correlative and require orthogonal validation. Finally, while the agentic workflow improves traceability, it introduces operational risks that still need human oversight (e.g., User Interface actions, dependency drift).

Our framework may positively impact precision medicine by enabling reproducible, interpretable integration of single-cell imaging with protein networks, improving biomarker discovery and drug mechanism studies. Potential risks include bias in PPI resources, over-interpretation of correlative enrichments, dual-use concerns, and environmental costs of computation. These can be mitigated by open reproducible practices, human oversight of LLM-assisted steps, orthogonal biological validation, responsible licensing, and reporting compute usage.

# 5    Conclusion

In this study, we developed and applied the Modified CellMaps pipeline that integrates object-level microscopy embeddings with PPI embeddings to generate biologically interpretable hierarchical maps. By combining segmentation, ViT feature extraction, and molecular network embeddings, the workflow preserved the biological consistency of the original CellMaps approach while improving resolution and specificity.

Looking ahead, further improvements can be achieved by adopting larger or domain-adapted ViT models, incorporating additional omics layers (e.g., transcriptomics and phosphoproteomics), and conducting orthogonal experimental validation.

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

# Author contributions statement

We used LLMs (ChatGPT 5, Gemini 2.5) as a co-pilot for planning, code drafting, and as an interactive agent to launch segmentation. AI contributed to segmentation (LLM-guided detection and mask generation), image feature extraction, and manuscript preparation (LaTeX structure and refinement). All code was reviewed and executed by the authors on HPC, outputs were validated (e.g., overlap statistics, GO/KEGG checks), biological interpretations were made by the authors, and the human team assumes full responsibility for the results.

# Competing interests

The authors declare that they have no conflicts of interest.

## Responsible AI Statement

This work adheres to the NeurIPS Code of Ethics and the Responsible AI requirements of Agents4Science. The AI system served as the primary contributor, generating code, drafting manuscript sections, and executing experimental pipelines under human oversight.

We recognize potential risks, including (i) bias or incompleteness in public protein–protein interaction resources, (ii) over-interpretation of agent-generated outputs without biological validation, (iii) dual-use concerns in applying automated pipelines to sensitive biomedical data, and (iv) environmental impact from compute requirements. Mitigation strategies include transparency of AI involvement, open release of anonymized code and reproducible workflows, explicit human validation of outputs, reporting of compute usage, and limiting analyses to publicly available, non-identifiable datasets.

The anticipated broader impacts include advancing reproducible multimodal biomedical AI, accelerating discovery through interpretable agentic workflows, and lowering technical barriers for interdisciplinary researchers while ensuring safe and responsible deployment.

## Reproducibility Statement

We provide all materials required to reproduce our results.

**Code and Artifacts.** An anonymized repository with source code, experiment scripts, and run logs is provided at [ANONYMIZED_REPO_URL].

**Data Access.** We use publicly available CM4AI resources (subset described in the paper: immunofluorescent microscopy for untreated, paclitaxel, and vorinostat conditions). The repo includes a downloaded version of the dataset.

**Environment and Determinism.** Experiments were executed on an HPC cluster with NVIDIA GPUs. We fix random seeds across `numpy`, `torch`, and Python (`seed=42`).

**Preprocessing and Pipelines.** The repository provides callable pipelines for (1) cell/object detection and mask generation, (2) ViT-based single-cell feature extraction, (3) PPI embedding generation and alignment, and (4) hierarchical analyses. Each stage has a corresponding Jupyter notebooks.

**Hyperparameters and Evaluation.** Default hyperparameters are reported in the paper.

**Compute Reporting.** For transparency, we log wall-clock time, GPU model/count in the paper. This enables independent estimation of compute and energy cost.

**Licensing and Usage.** Code is released under a permissive license compatible with the dataset terms (see `LICENSE` and dataset licenses referenced in the README).

## Additional information

The source code is made public via the link https://anonymous.4open.science/r/CM4AI-56A4/README.md

## A    Technical Appendices and Supplementary Material

Technical appendices with additional results, figures, graphs and proofs may be submitted with the paper submission before the full submission deadline, or as a separate PDF in the ZIP file below before the supplementary material deadline. There is no page limit for the technical appendices.

## Agents4Science AI Involvement Checklist

1. **Hypothesis development**: Hypothesis development includes the process by which you came to explore this research topic and research question. This can involve the background research performed by either researchers or by AI. This can also involve whether the idea was proposed by researchers or by AI.

    Answer: B — AI-led ideation with human oversight/selection.

    Explanation: ChatGPT proposed stepwise pipelines and brainstormed directions; humans evaluated options, refined scope, and chose what to implement. AI organized ideas but did not make autonomous decisions.

2. **Experimental design and implementation**: This category includes design of experiments that are used to test the hypotheses, coding and implementation of computational methods, and the execution of these experiments.

    Answer: C — Mixed: AI-generated code & human-run execution.

    Explanation: The LLM drafted segmentation and processing scripts and helped plan experiments; one agent operated mostly automatically, while the other tasks were executed by humans on the HPC, who handled failures, tuned parameters, and integrated outputs.

3. **Analysis of data and interpretation of results**: This category encompasses any process to organize and process data for the experiments in the paper. It also includes interpretations of the results of the study.

    Answer: B — AI-led interpretation with human oversight.

    Explanation: About 90% of the analysis and interpretation was performed with LLM support. ChatGPT and Gemini organized, summarized, and contextualized the data outputs (e.g., pathway enrichments, cluster comparisons, visualizations), producing first-pass interpretations. Humans reviewed, corrected possible hallucinations, and finalized the biological narratives.

4. **Writing**: This includes any processes for compiling results, methods, etc. into the final paper form. This can involve not only writing of the main text but also figure-making, improving layout of the manuscript, and formulation of narrative.

    Answer: B — AI-led drafting with human oversight/edits.

    Explanation: The LLM produced drafts, LaTeX structure, and figure captions; humans verified accuracy, corrected errors, and finalized the narrative.

5. **Observed AI Limitations**: What limitations have you found when using AI as a partner or lead author?

    Description:

    **Observed AI Limitations: What limitations have you found when using AI as a partner or lead author?**

    - **Task execution.** Because our pipeline runs in Jupyter notebooks, many steps require interaction with a web interface. The ChatGPT-provided agent performs well when instructions are precise and can attempt to correct errors, though some issues may persist and still require human intervention.
    - **Idea generation.** ChatGPT is a strong brainstorming assistant; some ideas are genuinely valuable, but novelty and feasibility still require human vetting.
    - **Manuscript writing.** Agents can deliver rigorous "harsh reviews" (via LLM API calls) and, when combined, enable fast revision cycles. However, factual accuracy and citation integrity must be checked by humans.
    - **Interpretation of results.** AI can produce high-quality summaries and explanations—especially for non-experts—but domain-specific nuances and causal claims should be validated by human experts.

    **Overall.** Multi-agent LLM workflows are promising and can accelerate research, but they require careful oversight, verification, and error handling to be reliable at scale.

