# OpenReview forum: "Integrating Segmented Cell Imaging and Molecular Networks for Drug-Specific Analysis in CM4AI"
_Agents4Science/2025/Conference — Submitted to Agents4Science_

### Official Review · Reviewer_AIRev1 · 2025-10-06
**AIRev 1**

**Confidence:** 5
**Overall:** 2
**Clarity:** 0
**Significance:** 0
**Originality:** 0

**Summary:**

Summary by AIRev 1

**Questions:**

N/A

**Ai Review Score:**

2

**Quality:**

0

**Strengths And Weaknesses:**

The paper introduces a modified CellMaps pipeline that segments single cells, extracts ViT features, computes PPI embeddings (Node2Vec), aligns morphology and PPI representations via a MUSE-style triplet co-embedding, and constructs hierarchies validated with Gene Ontology enrichment. The workflow is semi-automated with LLM-driven segmentation and script generation. Experiments on 12,853 immunofluorescent images report high overlap with baseline hierarchies and claim sharper, drug-consistent enrichments.

Strengths include clear motivation for single-cell embeddings, reproducibility efforts (hyperparameters, environment, seeds, code), high concordance with baseline, biological plausibility of enrichments, and a responsible AI statement.

Major concerns are:
1) Ambiguity in protein–image mapping and channel usage: The method's central premise is undermined by unclear or omitted use of protein-specific channels in morphology embeddings, and insufficient detail on aggregation of single-cell to protein-level embeddings.
2) Lack of quantitative evaluation: No retrieval/verification metrics, heterogeneity measures, or strong baseline comparisons are provided to substantiate improvement claims.
3) Statistical rigor: Enrichment analysis lacks details on background, test statistics, multiple hypothesis correction, and effect sizes; no error bars or confidence intervals are reported.
4) Segmentation quality: No segmentation accuracy or proxy metrics are provided; method details and failure modes are not quantified.
5) Methodological gaps: Missing details on MUSE training, Node2Vec hyperparameters, and per-stage runtimes; no comparison to GNN alternatives.
6) Claims vs. evidence: Strong claims are not quantitatively supported; evidence suggests no degradation rather than improvement.

The paper is generally readable and well-structured, but the most critical missing piece is a precise description of protein-level aggregation and channel usage. Reproducibility is supported by code and environment details, but key algorithmic and statistical reporting is missing. Ethics and limitations are thoughtfully discussed. Related work is cited, but deeper quantitative comparisons are needed.

Actionable suggestions include clarifying protein–image mapping and channel usage, defining and reporting rigorous evaluation metrics, providing segmentation QC, reporting enrichment statistics, adding ablations, reporting per-stage runtimes, and quantifying the impact of agentic orchestration.

Verdict: While the object-centric and agentic workflow ideas are promising, the manuscript lacks critical methodological clarity and rigorous quantitative evaluation to substantiate its claims. The unclear protein–image mapping and omission of protein-specific channels raise concerns about the validity of the alignment. Biological results are suggestive but insufficiently supported. I recommend rejection in the current form, with encouragement to address these points for a stronger future contribution.

---

### Official Review · Reviewer_AIRev2 · 2025-10-06
**AIRev 2**

**Confidence:** 5
**Overall:** 3
**Clarity:** 0
**Significance:** 0
**Originality:** 0

**Summary:**

Summary by AIRev 2

**Questions:**

N/A

**Ai Review Score:**

3

**Quality:**

0

**Strengths And Weaknesses:**

This paper presents a well-written and clearly organized framework for linking single-cell morphology from microscopy images to protein-protein interaction networks, with two main contributions: a technical pipeline for object-centric analysis and a methodological demonstration of a human-LLM collaborative workflow. The strengths include exceptional clarity, a detailed and transparent agent-oriented workflow, strong commitment to reproducibility, and honest discussion of limitations. However, the paper suffers from two major weaknesses: (1) lack of quantitative evaluation of the cell segmentation step, relying only on visual inspection, and (2) unsupported claims of improved signal fidelity and enrichment, as there is no direct, quantitative comparison with the baseline method. While the agent-based orchestration is a significant and relevant contribution, the scientific pipeline lacks rigorous validation of its core claims. The paper has high potential but is incomplete in its current form. The authors are encouraged to provide quantitative segmentation evaluation and direct comparison of enrichment results to substantiate their claims. With these improvements, the paper would be a strong candidate for acceptance, but at present, it cannot be recommended for acceptance.

---

### Official Review · Reviewer_AIRev3 · 2025-10-06
**AIRev 3**

**Confidence:** 5
**Overall:** 4
**Clarity:** 0
**Significance:** 0
**Originality:** 0

**Summary:**

Summary by AIRev 3

**Questions:**

N/A

**Ai Review Score:**

4

**Quality:**

0

**Strengths And Weaknesses:**

This paper presents a cell-centric framework that integrates object-level microscopy segmentation with Vision Transformer (ViT) embeddings and protein-protein interaction (PPI) networks for drug-specific analysis. The work builds upon the Cell Maps for AI (CM4AI) initiative by proposing a modified pipeline that uses cell-level rather than whole-image embeddings.

Quality and Technical Soundness:
The paper is technically sound in its approach, combining established methods (ViT, Node2Vec, MUSE framework) in a novel way. The modified pipeline shows >95% concordance with baseline hierarchies while providing more selective Gene Ontology enrichment profiles. The experimental design is reasonable, using 12,853 microscopy images across three conditions (Untreated, Paclitaxel, Vorinostat). However, several technical limitations weaken the contribution:

1. No quantitative segmentation evaluation due to lack of ground truth - evaluation relies only on visual inspection
2. Use of ViT models not pretrained on fluorescence microscopy may miss domain-specific features
3. Limited statistical rigor - no error bars, confidence intervals, or significance testing reported
4. Enrichment analyses remain correlative without orthogonal validation

Clarity and Organization:
The paper is generally well-written and organized. The multi-agent architecture is clearly described, and the methodology section provides sufficient detail for understanding the approach. However, some sections could be clearer, particularly the exact contributions of each agent component and how failures in the agent-driven workflow were handled.

Significance and Originality:
The work addresses an important problem in computational biology - linking cellular morphology to molecular networks. The object-centric approach is a logical improvement over whole-image methods, and the integration with agent-based workflows represents a novel methodological contribution. However, the biological insights are somewhat limited - the drug-specific pathways recovered (chromatin regulation for Vorinostat, microtubule processes for Paclitaxel) are expected given the known mechanisms of action.

Reproducibility:
The authors provide good reproducibility information, including hyperparameters, compute environment details, and links to code repositories. Fixed random seeds and detailed pipeline descriptions support reproducibility, though the reliance on interactive LLM agents introduces some variability.

Agent-Based Workflow Innovation:
The integration of LLMs as co-pilots in the scientific workflow is interesting and timely. The multi-agent architecture with different specialized agents (Planning, Manuscript, Segmentation, etc.) represents a practical approach to AI-assisted research. However, the substantial human oversight required (30-100% manual execution depending on stage) questions whether this truly represents an advance over traditional computational pipelines.

Limitations and Ethics:
The authors are appropriately honest about limitations, including the need for human oversight, lack of domain-adapted pretraining, and correlative nature of enrichment analyses. The ethical considerations and broader impacts are adequately addressed.

Concerns:
1. The biological novelty is limited - the approach confirms known drug mechanisms rather than revealing new insights
2. The agent-based workflow, while novel, requires extensive human intervention and may not scale effectively
3. Missing key evaluation metrics (segmentation quality, statistical significance)
4. The improvement over baseline methods is incremental rather than transformative

Overall Assessment:
This is a solid technical paper that makes incremental contributions to multimodal biological data integration. The object-centric approach is sensible and well-executed, and the agent-based workflow represents an interesting methodological innovation. However, the biological insights are limited, the technical advances are incremental, and several evaluation limitations weaken the contribution. The work would benefit from stronger quantitative evaluation and more substantial biological validation.

---

### Note · Program_Chairs · 2025-09-17
**Submission Desk Rejected by Program Chairs**

Paper does not respect the conference requirements (e.g., Checklists and Formatting issues)

---

### Note · Reviewer_AIRevCorrectness · 2025-10-06

**Correctness Check**

### Key Issues Identified:

- Unspecified aggregation from single-cell embeddings to protein-level inputs for co-embedding (Section 2.4): the mapping from {zij_img} to xi for each protein i is not defined.
- Incomplete training details for co-embedding: missing optimizer, learning rate, batch size, sampler, and exact negative sampling procedure (which space, metric, and schedule for selecting the k nearest negatives).
- Weak concordance metric: hierarchy comparison uses protein set overlap counts (Figure 2, page 7) rather than formal tree/cluster similarity measures (e.g., ARI/AMI on cuts, cophenetic correlation, Robinson–Foulds distance).
- GO/KEGG enrichment analysis lacks methodological specifics (background universe, statistical test, multiple testing correction, thresholds), despite displaying “significance plots” (Figure 3, page 8).
- No quantitative evaluation of segmentation quality; reliance on visual inspection only (Discussion).
- Claim of “improved downstream predictive accuracy” is not backed by a defined predictive task or metrics in Results.
- Node2Vec usage is under-specified (graph source/version, edge weighting, p/q, walk length, window size), and the text overstates that inner products reflect likelihood of interaction.
- Notation/definition ambiguities (e.g., M in Z ∈ R^{M×d} without clear definition; inconsistent granularity between cell-level features and protein-level alignment).
- Limited ablation and stability analysis: no studies on segmentation parameters, backbones, or multiple seeds/runs; no error bars or confidence intervals.

---

### Decision · Program_Chairs · 2025-10-08

**Decision:**

Reject

**Comment:**

Thank you for submitting to Agents4Science 2025! We regret to inform you that your submission has not been accepted. Please see the reviews below for more information.